# Charge Carrier Relaxation in Colloidal FAPbI_3_ Nanostructures Using Global Analysis

**DOI:** 10.3390/nano10101897

**Published:** 2020-09-23

**Authors:** Carolina Villamil Franco, Benoît Mahler, Christian Cornaggia, Thomas Gustavsson, Elsa Cassette

**Affiliations:** 1Université Paris-Saclay, CEA, CNRS, Laboratoire Interactions, Dynamiques et Lasers (LIDYL), 91191 Gif-sur-Yvette, France; carolina.villamil-franco@cea.fr (C.V.F.); christian.cornaggia@cea.fr (C.C.); thomas.gustavsson@cea.fr (T.G.); 2Université de Lyon, Université Claude Bernard Lyon 1, CNRS, Institut Lumière Matière (ILM), F-69622 Villeurbanne, France; benoit.mahler@univ-lyon1.fr

**Keywords:** hot charge carrier relaxation, Auger recombination, colloidal perovskite nanocrystals, transient absorption spectroscopy

## Abstract

We study the hot charge carrier relaxation process in weakly confined hybrid lead iodide perovskite colloidal nanostructures, FAPbI_3_ (FA = formaminidium), using femtosecond transient absorption (TA). We compare the conventional analysis method based on the extraction of the carrier temperature (*T_c_*) by fitting the high-energy tail of the band-edge bleach with a global analysis method modeling the continuous evolution of the spectral lineshape in time using a simple sequential kinetic model. This practical approach results in a more accurate way to determine the charge carrier relaxation dynamics. At high excitation fluence (density of charge carriers above 10^18^ cm^−3^), the cooling time increases up to almost 1 ps in thick nanoplates (NPs) and cubic nanocrystals (NCs), indicating the hot phonon bottleneck effect. Furthermore, Auger heating resulting from the multi-charge carrier recombination process slows down the relaxation even further to tens and hundreds of picoseconds. These two processes could only be well disentangled by analyzing simultaneously the spectral lineshape and amplitude evolution.

## 1. Introduction

Thanks to their outstanding properties, lead halide perovskites have emerged as extremely promising low-cost processing materials for several optoelectronic applications as solar cells [1,2], photo-detectors [3,4], light-emitting diodes [5,6] and lasers [7,8]. For all these applications, it is of main importance to characterize the rate at which hot charge carriers relax to the band-edge (cooling). For example, slow charge carrier cooling is advantageous in single junction solar cells where the extraction of the hot charge carriers could overpass the Shockley-Queisser limit [5]. In particular, the confinement effect in perovskite nanostructures could potentially efficiently slow down the cooling process by orders of magnitude through the intrinsic phonon bottleneck effect [6].

The most widely spread technique to investigate hot charge carrier relaxation in perovskite materials is transient absorption (TA) spectroscopy [7,8,9]. The high-energy tail of the band-edge bleach resulting from the Burstein-Moss effect reflects the population of the continuous high-energy levels above the bandgap and can be described by a Fermi-Dirac distribution with a characteristic carrier temperature, *T_c_*. The investigation of the full cooling dynamics using this conventional method is based on the extraction of a large number of *T_c_* by tail fitting of the TA spectra over a wide range of times *t* (several orders of magnitude). Typical fs-TA spectroscopy allows to access the relaxation dynamics from the sub-ps timescales to several nanoseconds with a well-defined excitation fluence and energy. However, the obtained *T_c_* values from this physically motivated approach, are not necessarily accurate. In particular, they have been found to be strongly dependent on the energy range used for the tail fitting [9,10]. For instance, in the region of interest, the bleach signal will be more or less superposed with the large broadband photo-induced absorption signal at energies above the band-edge [11], which can affect the fit. Moreover, nanocrystal samples are susceptible to present inhomogeneous spectral lineshapes due to size dispersion, which can artificially induce higher carrier temperatures. All of those imprecisions in the determination of *T_c_* will have a direct impact on the energy-loss rate values, proportional to d*T_c_*/d*t*, used to compare one sample to another in terms of composition or confinement effects [9]. 

In spite of such concerns, most of the reports use this conventional approach to extract carrier cooling dynamics not only in bulk hybrid halide perovskite materials (thin films) [7,8,12], but also in halide perovskite nanocrystals (NCs) [10,12,13]. In all these cases, this method can be applied as long as the energy level spacing remains below the thermal energy. In such polar semiconductors, the relaxation is governed by fast carrier-optical phonon scattering [14]. Reported carrier cooling times range from 210 to 600 fs in MAPbI_3_ (MA = methylammonium) thin films [7] or from 100 to 800 fs in MAPbBr_3_ thin films [13] with carrier densities from about 10^17^ to 10^19^ cm^−3^ and a similar excess of energy. This increase in cooling times with the initial excitation fluence is known as the hot phonon-bottleneck effect [15]. It is worth pointing out that in such bulk perovskite materials or weakly confined nanocrystals, in particular for iodide-based materials, the main photo-generated species are free carriers and not excitons [13,14,15]. In more strongly confined perovskite nanostructures, the conventional approach extracting *T_c_* is invalid. Alternatively, researchers use the buildup of the bandedge bleach over time to follow the cooling dynamics [16,17]. However, single wavelength trace analysis can be strongly complicated by excitonic effects such as Stark effects or coupled optical transition [18,19].

For weakly confined MAPbBr_3_ NCs, slightly slower carrier cooling has been observed than in the bulk counterpart when excited at 400 nm while not much difference could be observed when comparing the energy-loss rate [13]. In a subsequent work, the cooling times at the same excitation wavelength and low carrier density (~10^17^ cm^−3^) were found to be slightly dependent on the composition of lead bromide perovskites NCs with different cations, Cs, MA and FA: 310, 235 and 180 fs, respectively [16]. At high carrier densities (~10^19^ cm^−3^), the decay of *T_c_* with *t* presents an additional component on a time range order of magnitude higher: 5 ps for CsPbBr_3_ and 3 ps for MAPbBr_3_ and FAPbBr_3_ NCs [16] and even 10–30 ps in MAPbBr_3_ NCs [13]. The picosecond component was also reported in FAPbI_3_ NCs, together with another in a few hundreds of ps [10] that could not be observed in the shorter time range experiments of the previous authors. These substantially longer cooling times can be understood in terms of enhanced multi-particle processes in confined systems such as Auger recombination that leads to a re-heating effect [13,20] and thus should further increase the cooling times.

Here, we present a study of hot carrier relaxation dynamics in weakly confined FAPbI_3_ nanoplates (NPs) and nanocrystals (NCs) using femtosecond transient absorption spectroscopy at different excitation fluencies and photon energies. Special attention is given to the way of extracting carrier cooling times during the first few picoseconds. We compare the conventional method, i.e., the fitting of the high energy tail of the bandedge bleach, with a global analysis method using singular value decomposition (SVD). This latter method allows us to link the evolution in time of the charge carrier temperature and population (density). Additionally, the versatility of the global method allows us to extend the cooling analysis to a few nanoseconds, therefore covering then three distinct characteristic time regimes.

## 2. Materials and Methods 

### 2.1. Material Synthesis and Characterization

Colloidal FAPbI_3_ perovskite nanocrystals (NCs) and thick nanoplates (NPs) were synthesized by employing two different synthetic approaches. Cubic-shaped NCs were synthesized following a protocol based on the “hot-injection” (HI) crystallization method [21]. Thick FAPbI_3_ nanoplates were synthesized at room temperature based on the “ligand-assisted re-precipitation” (LARP) method [22]. The shape and size of the prepared colloidal perovskite nanostructures were determined by transmission electronic microscopy (TEM) using a JEOL 2100 equipped with a LaB_6_ filament and operating at 200 kV. Detailed synthetic methods are described in the Appendix A with the list of chemical used, from Sigma Aldrich, Alfa Aesar and Acros Organics.

### 2.2. Steady-State Spectroscopy

Absorption measurements were carried out with a UV/Vis Lambda 850 spectrophotometer (Perkin Elmer) covering the 175–900 nm spectral range. Photoluminescence measurements were carried out with a Fluorolog 3–22 spectrofluorometer (HORIBA JOBIN-YVON, Chilly Mazarin, France) equipped with a R928P photomultiplier tube detector (200–870 nm, Hamamatsu, Massy, France) and a continuous wave xenon arc lamp (450 W, 250–2500 nm). An ultraviolet enhanced silicon photodiode reference detector monitored and compensated for variation in the xenon lamp intensity. 

### 2.3. Time-Resolved Spectroscopy

Femtosecond transient absorption (TA) experiments were performed at room temperature using a home-built setup. Briefly, this uses a fundamental beam from an amplified Ti:Sa laser source (3 kHz, 40 fs, 2 W) split into two parts, one to generate the tunable excitation pump pulses from a home-made visible non-collinear optical parametric amplifier (NOPA) and the other part to produce the delayed white light continuum probe pulses. A more detailed description can be found in the Appendix A. Different excitation wavelengths were used, 630 nm (1.97 eV) and 520 nm (2.38 eV) for the NPs and 650 nm (1.91 eV) for the cubic NCs. The excitation fluence was chosen between 6 and 60 μJ/cm^2^ depending on the experiment. The maximum pump-probe time delay that could be reached was about 3.2 ns and the temporal resolution was about 130 fs, estimated from cross-phase modulation. Post-acquisition data treatments are also described in the Appendix A (e.g. chirp correction). Experiments were performed in transmission at room-temperature in a 1-mm thick circulating cell connected to a peristaltic pump to prevent photo-degradation at the focus position and photo-charging effects. The colloidal NCs and NPs were dispersed in anhydrous toluene or chloroform, with an optical density of less than 0.3 at the excitation wavelength and above.

### 2.4. Data Analysis

#### 2.4.1. Conventional Method: Tail-Fitting

The high-energy tail reflects the distribution of the thermalized charge carriers occupying the quasi continuous high-energy levels as well, since the energy spacing remains small in comparison with $k_{B}T$ at room temperature in our weakly confined nanostructures [10,13]. This hot population follows a Fermi-Dirac distribution, which can be approximated by a Maxwell-Boltzmann function [15]:(1)$$\Delta A\left(h\nu \right)=A_{0}exp\left(-\frac{h\nu -E_{eff}}{k_{B}T_{c}}\right)+PIA$$
where, *∆A* is the transient absorption signal, *A*_0_ is a constant, *hν* is the detected energy, *k*_B_ is the Boltzmann constant and *T_c_* is the charge carrier temperature. The PIA corresponds to the broadband photo-induced absorption signal. *E_eff_* corresponds to the time-dependent center energy of the bandedge bleach signal due to the Burstein-Mott shift.

#### 2.4.2. Global Analysis

The relaxation dynamics were accessed using a global analysis method based on singular value decomposition (SVD) to follow the evolution in time of the complex TA signal with several overlapping spectral components [16,23,24]. The open-source software Glotaran [25] was used to perform a global analysis of the TA data. The sequential kinetic model, with spectral components known as Evolution Associated Spectra (*EAS_i_*) and related by successively decreasing rate constants (*k*_i_ = 1/*τ*_i_), is given by the following equation:(2)$$\Delta A\left(\lambda ,t\right)=\overset{n}{\underset{i=1}{\sum }}\left(M_{i}EAS_{i}\left(\lambda \right)-M_{i-1}EAS_{i}\left(\lambda \right)\right)$$
with the associated time-dependent amplitude *M*_i_(t):$$M_{0}=0$$
(3)$$M_{i}\left(t\right)=\frac{1}{2}exp\left(-k_{i}t\right)exp\left[k_{i}\left(t_{0}+\frac{k_{i}{\tilde{\Delta }}^{2}}{2}\right)\right]\left[1+erf\left[\frac{t-\left(t_{0}+k_{i}{\tilde{\Delta }}^{2}\right)}{\sqrt{2}\tilde{\Delta }}\right]\right]\;for\;i>0$$
assuming a Gaussian instrument response function (IRF), with full width at half maximum *Δ* and centered at *t*_0_ (time zero of the pump probe overlap) [26]. We have $\tilde{\Delta }=\Delta /\left(2\sqrt{ln\left(2\right)}\right)$. In this first-order approximation, *M*_i_ is a mono-exponential decay function (first term) that is convoluted with the IRF (giving the second term). The last term is a step function, more or less smooth depending on *Δ*, to generate the signal starting around *t*_0_.

Here, the *EAS_i_* represent the different spectral contributions of the system state. This independent analysis of spectral and temporal characteristics facilitates the identification of the corresponding processes. In the simplest case with only an initial (*EAS_i_*) and a final (*EAS_f_*) state during the first relaxation stage, the time-dependent carrier temperatures can be extracted from the corresponding spectral components assuming a thermalized distribution of the charge carriers. Furthermore, the time constant *τ*_1_ = 1/*k*_1_ obtained from the global data analysis can be used to simulate the evolution of the carrier temperature *T_c_* using the following equation:(4)$$T_{c}\left(t\right)=\;\left(T_{i}-T_{f}\right)\;exp\left(-\frac{t}{\tau_{1}}\right)+T_{f}$$
where *T_i_* and *T_f_* are the initial and final carrier temperatures obtained from the *EAS_i_* and *EAS_f_* by the conventional method explained above.

## 3. Results

### 3.1. Sample Characterization

The steady-state absorption and emission spectra of the cubic-shaped FAPbI_3_ nanocrystals (NCs) and the thick FAPbI_3_ nanoplates (NPs) are shown in Figure 1a. The absorption spectra of both samples extend to the near-infrared and present no well-defined excitonic structure, as expected in these weakly confined materials characterized by a small exciton binding energy. In the absence of a well-defined feature in the absorption spectrum of the two samples, the bandgap was estimated from the central energy of the photo-induced bandedge bleach of the TA data on a long time scale (away from the Burstein-Mott shift effects). It is 1.66 eV (745 nm) for the cubic-shaped NCs and 1.62 eV (765 nm) for the thick NPs. The cubic NCs exhibit a photoluminescence (PL) peak centered around 1.65 eV (752 nm) while the PL maximum of the thick nanoplates is at 1.61 eV (770 nm). The blue-shifted PL suggests some degree of confinement when compared with the typical 820–840 nm PL emission of bulk FAPbI_3_ thin films [27] and macrocrystals [28]. 

The NP thickness is similar to the average size of the cubic NCs, while the optical spectrum is slightly red-shifted in comparison due to the larger lateral dimensions of the NPs. The histograms of the measured sizes (extracted using the ImageJ program) are displayed in Figure 1b with their representative TEM pictures. The average size for the cubic FAPbI_3_ NCs is 12 ± 2 nm while the NP lateral dimensions were found to be 70 ± 20 nm and their thickness about 11 nm (Figure 1b, inset). This average thickness corresponds to approximatively 18 monolayers considering a thickness along the < 100 > crystallographic direction (0.6 nm each). The sizes and linear spectra of both samples are in agreement with a weak confinement when compared with the ≈5 nm exciton Bohr radius reported for FAPbI_3_ [29].

### 3.2. Carrier Relaxation Dynamics

In order to make sure to investigate the full cooling dynamics, we performed femtosecond TA experiments over five decades of times, from hundreds of femtoseconds up to a few nanoseconds. The TA spectra during the first 3 ps of the thick colloidal FAPbI_3_ nanoplates (NPs) excited at 1.97 eV (630 nm) are shown in Figure 2a. This excitation above the bandgap corresponds to an excess energy of about 350 meV. The TA spectra present the same characteristic features as hybrid perovskite thin films [30]: (1) a large bleach signal at about 1.7 eV corresponding to band-edge state filling and extending to higher energies at early times (Burstein-Moss shift), (2) a second bleach signal at higher energy (>2.6 eV, out of scale) involving a higher energy transition and (3) a broad photo-induced absorption (PIA). It should be noted that the sign of the TA spectra in Figure 2a was inversed before normalization so that negative features appeared positive and inversely. As discussed in the Introduction, the high-energy tail reflects the distribution of the thermalized charge carriers occupying the quasi-continuous high-energy levels, since the energy spacing in these nanostructures remains small in comparison with $k_{B}T$ at room temperature [10,13].

Using Equation (1) to fit the high-energy tail (1.9–2.1 eV) of the band-edge bleach of each TA spectrum (see for example Figure 2a), time-dependent carrier temperatures were extracted as described in the Materials and Methods section. Typical *T_c_* curves obtained for moderate- (6 μJ/cm^2^) and high-excitation fluence (60 μJ/cm^2^), ranging from 300 fs to 3 ns, are displayed in Figure 2b for FAPbI_3_ NPs. The average densities of electron-hole pairs created per pulse for these two fluences are estimated to be 5.6 × 10^17^ and 5.6 × 10^18^ cm^−3^, respectively (see calculation in Appendix A). These curves can be fitted satisfactorily using a multi-exponential decay function (R^2^ factor 0.924 and 0.984, respectively), with the resulting parameters given in Table 1.

We first focused on the short picosecond time-range (0–3 ps), where the amplitude of the associated time constant represents 90% of the total relaxation in terms of temperature decay at moderate excitation fluence and 70% at high fluence, as it is shown for the amplitudes of the tri-exponential fit in Table 1. As outlined in the Introduction, the extraction of the carrier temperature by the tail-fitting method can be relatively inaccurate so we turned to the global analysis (see Materials and Methods section above) in order to follow the evolution of the full lineshape in time. For that, we used two spectral components EAS_1_ and EAS_2_, with corresponding kinetic rates *k*_1_ and *k*_2_, such that EAS_1_ rises during the interaction with the pump pulse and decays exponentially with *k*_1_, while EAS_2_ rises with *k*_1_ and then decays exponentially with *k*_2_ (see Equations (2) and (3)). This approach satisfactorily reproduces the experimental TA data for all the excitation fluences as can be seen in selected decay traces in Figure A1c,d in Appendix B. Moreover, no significant improvement was obtained in the Glotaran root mean square (RMS) deviation when using more than two components for the global fit.

An example of the EAS spectral components obtained by this sequential model is given for high fluence (60 µJ/cm^2^) in Figure 3a. The spectral components at different excitation fluences are displayed in Appendix B. It should be noted that during the first picosecond time range, EAS_1_ evolves into EAS_2_, presenting a much reduced high-energy tail.

A comparison between the carrier temperature decays for FAPbI_3_ NPs obtained by the conventional high-energy tail fitting method (dots) and the global analysis (lines) is shown in Figure 3b. The resulting kinetic parameters from the global analysis are given in Table 2. We consider the evolution from EAS_1_ to EAS_2_ with the rate constant *k*_1_ as the early-stage relaxation of the hot charge carriers to the band-edge (hot carrier cooling). The resulting associated lifetime *τ*_1_ increases with the excitation fluence from 360 ± 20 fs at 6 µJ/cm^2^ to 970 ± 80 fs at 60 µJ/cm^2^, which is, as we will discuss later, typical of the hot phonon bottleneck effect.

In order to study the influence of the excitation photon energy on the relaxation dynamics of the FAPbI_3_ nanostructures, we performed similar experiments by exciting the NPs at higher photon energy (520 nm, excess energy of about 760 meV). Moreover, to see the influence of the morphology and/or surface ligands on this early stage of relaxation, additional measurements were performed on cubic-shaped NCs excited at 650 nm (about 240 meV above the bandgap energy of 1.67 eV). The experimental data were analyzed using the global analysis method described above. All the relaxation *k*_1_ rates obtained at different excitation fluences are plotted in Figure 4a.

At the lowest fluences, the cooling is relatively fast and depends on the excitation photon energy: *τ*_1_ = 1/*k*_1_ ≈ 411 ± 1 fs for the thick FAPbI_3_ nanoplates excited at 520 nm and 350 ± 10 fs when excited at 630 nm. While physically it takes a longer time to fully relax to the band-edge in the case of a higher excess energy, the “global energy loss rate” defined as the initial excess energy (Δ*E* = h*υ* − *E*_g_) divided by the cooling time (*τ*_1_) is 1.9 and 1.0 eV/ps, respectively (Figure 4b). It is thus higher when it is excited at 520 nm than at 630 nm, as previously reported in bulk thin film perovskites [14]. For the cubic-shaped FAPbI_3_ NCs excited at 650 nm, we obtained *τ*_1_ ≈ 240 fs, corresponding to the global energy-loss rate of about 0.99 eV ps^−1^. At higher carrier densities, the initial cooling time *τ*_1_ increases up to almost one picosecond for the two FAPbI_3_ samples, similarly to previously reported value for FAPbI_3_ NCs [10]. This corresponds to an effective energy-loss rate of about 0.33 eV ps^−1^ for the NPs excited at 630 nm.

Returning to the thick NPs excited at 630 nm, after applying the global analysis method to study the early stage of the cooling dynamics in the short-time range, we applied it to longer time ranges. Indeed, the carrier temperature still evolves over tens and hundreds of ps, in particular at high excitation fluence (cf Figure 2b). In order to cover the full nanosecond time range with appropriate time steps, we divided the experiments into a middle time range, up to 200 ps with a step of 1 ps and a long time range, up to 3.2 ns with steps of 20 ps. 

For these two longer time ranges, three kinetic components were needed to reproduce the data well (the Glotaran RMS deviations were significantly lower for three components than for two). The three EAS components at high excitation fluence (60 μJ/cm^2^) are shown in Figure 5a for the middle range up to 200 ps and in Figure 5b for the long time range up to 3.2 ns. The experimental decay traces at different excitation fluences are shown in Figure 5c,d along with the resulting fits. For comparison, the EAS obtained at moderate excitation fluence (15 μJ/cm^2^) are displayed in Appendix B. All the time constants obtained from the global analysis in the middle time range (*τ*_i,m_) and in the long time range (*τ*_i,l_) for different excitation fluences are given in Table 3.

For the middle time range analysis, the characteristic time *τ*_1,*m*_ corresponding to the evolution from EAS_1_ to EAS_2_ ranges from sub-ps to a few ps and thus corresponds well to the initial stage of the relaxation discussed above. However, we should note that the time step of 1 ps does not allow to determine this time constant with accuracy, especially at low excitation fluence. On the other hand, the second and third time constants are about 25–40 ps (EAS_2_ to EAS_3_) and 280–510 ps (decay of EAS_3_), respectively. Here, a clear decrease with the excitation fluence can be observed in both time constants. 

In the case of the long-time range extending up to about 3 ns, the characteristic times are about 60–80 ps (EAS_1_ to EAS_2_) and 500–800 ps (EAS_2_ to EAS_3_). As in the analysis of the middle time range, the first time constant *τ*_1,l_ decreases with the excitation fluence. Curiously, an opposite trend was observed for the second time constant *τ*_2,l_. In addition, a time constant *τ*_3,l_ in the order of several of ns was obtained from the fit but since it is out of the time range we do not take it into account.

## 4. Discussion

### 4.1. Limitations of the High-Energy Tail Fitting Method

Although using the classical tail-fitting method to extract the carrier temperature *T_c_* from the TA spectra and then plotting the energy-loss rate versus this temperature allows to compare the cooling dynamics between different samples or, different initial excess energies for a given sample [13,14], we found it quite sensitive to the energy range used for the fit (Figure 2a). While some authors suggest a minimum range of 0.2 eV to ensure a mono-exponential decay to fit with Equation (1) [10], an extended energy range leads to an overlap of the band-edge bleach with the PIA signal that might introduce errors in the values of the extracted *T_c_*. Moreover, for a high density of photo-generated charge carriers, a non-negligible bandgap renormalization occurs, causing an important red-shift of the bleach [23]. This Coulomb screening effect has an opposite trend than the Burstein-Mott shift [31]. Even if the effective bandgap energy *E_eff_* is modified by this bandgap renormalization, its incidence on the *T_c_* value remains unclear. Finally, in the case of confined systems, the extracted carrier temperature can be artificially overestimated owing to a broad size-distribution. The high-energy tail of the PL spectrum (Figure 1a) can be understood as the presence of smaller-sized nanoplates produced by the “LARP” method. This can also result in a tail in the absorption band and thus can be confused with a higher temperature of the thick NPs. Experimentally, the final carrier temperature obtained for the NPs was higher than the expected lattice temperature at room conditions *T*~300 K (Figure 2b). 

In spite of this, the overall time dependence of the carrier temperature can be effectively described by a multi-exponential decay function, in line with previously reported relaxation dynamics in other FAPbI_3_ NCs [10]. In our case, the time constants reproduce the extracted *T_c_* values satisfactorily over the full nanosecond time range (Figure 2b). The resulting fit parameters presented in Table 1 correspond to a first component (*τ*_1_) from hundreds of femtoseconds to picoseconds, a second component (*τ*_2_) of tens of ps and a last component (*τ*_3_) of several hundreds of ps to ns. We will see that these characteristic time components will be retrieved in the global analysis when applied to different time ranges.

### 4.2. Global Analysis Versus High-Energy Tail Fitting over Short Times

Focusing on the time-evolution during the first few picoseconds, the evolution from a specific state, a “hot” state of the system, represented by EAS_1_, to another specific state, EAS_2_ can be considered as a phenomenological approximation to determine the cooling dynamics. In this sense, the system physically evolves through a continuum of states, each of them described by its own *T_c_*. However, as EAS_1_ and EAS_2_ overlap over an important spectral range (Figure 3a), a sequence of linear combinations of them produces an effective continuous shift of the band-edge bleach and a successive decrease in this high-energy tail over all time ranges of the experiment. Finally, the two resulting curves obtained by the tail-fitting method and the global analysis are in very good agreement during this first picosecond time range (see Figure 3b). 

The drawbacks of the conventional approach in terms of characteristic cooling time is absent in the global analysis method as it does not rely on the subjective tail fitting but results from a full spectral evolution analysis. The sequential model, where EAS_1_ decays with a characteristic rate *k*_1_ to the EAS_2_ state, gives a good agreement between the experimental and fitted data, as can be seen in the decay traces at the band-edge and in the PIA region, for all the excitation fluences (see Figure A1c,d). 

### 4.3. On the Hot Carrier Relaxation Mechanisms

Several mechanisms can contribute to the charge carrier relaxation and recombination dynamics in these weakly confined samples: the hot-charge carrier cooling through charge carrier-longitudinal optical (LO)-phonon scattering, Auger recombination (Auger heating) and electron-hole recombination. The associated processes can lead to different recombination orders (e.g. mono-, bi- or tri-molecular) depending on their physical nature. They will appear in different time ranges, as they are dominant under specific conditions such as the charge carrier density. 

### 4.4. LO-Phonon Scattering as the First Relaxation Stage

Using both conventional and global analysis methods, the fast sub-ps to ps component is strongly dependent on the excitation fluence. The characteristic time of this first decays increases from 0.25 to 2.1 ps with the tail-fitting and from 0.36 to 0.97 ps with the global analysis when going from 6 to 60 µJ/cm^2^. We attribute the evolution from EAS_1_ to EAS_2_, and the corresponding rate constant *k*_1_, to the early-stage relaxation of a hot charge carrier population to a much more relaxed one through the carrier—LO-phonon scattering, in agreement with previous work reported in bulk and bulk-like hybrid lead halide perovskites [7,8,9,14]. The slowdown of the relaxation at high charge carrier density is characteristic of the hot phonon bottleneck effect. Although the exact mechanism of the hot phonon bottleneck effect is still under debate [30], the optical phonon mode(s) involved can be assigned to the Pb-I inorganic lattice with no (or weak) contribution from the organic cation vibrational modes [32,33]. The good agreement between the experimental data and a simple mono-exponential decay in this short time range is rather surprising since a more complex *T_c_* evolution in time was expected [9,13,14].

### 4.5. Auger Recombination Further Slows Down the Relaxation in a Longer Time Range

In the tens to hundreds of picosecond time range, the high-energy tail of the bleach is still evolving, in particular at high excitation fluence, indicating that some additional slower mechanism is involved in the hot charge carrier relaxation. It can be seen by comparing EAS_1_ and EAS_2_ of the middle time range in Figure 5a that the total amplitude of the band-edge bleach does not evolve much during the first 10 ps. This means that the population (i.e., number of charge carriers) remains approximately constant during this initial stage of relaxation. The first time constant *τ*_1,m_ covering the time-range from sub-ps to a few picoseconds in this middle time range experiments corresponds well to the initial stage of relaxation described above by *τ*_1_ in the short time range. While the total areas of the EAS_1_ and EAS_2_ bleach signals in the middle time range experiments are rather similar, a strong diminution of the signal was observed during the second stage, covering several tens of ps. This can be observed by comparing EAS_2_ to EAS_3_ in the middle time range experiments associated with time constant *τ*_2,m_ (Figure 5a) or EAS_1_ to EAS_2_ in the long time range data associated with *τ*_1,*l*_ (Figure 5b).

At longer times, we found almost no evolution of the spectral lineshapes while the amplitude still evolves with characteristic times *τ*_2,*l*_ of a few hundreds of picoseconds and with *τ*_3,*l*_ of several nanoseconds (Figure 5b, inset). The change in time of the bleach amplitude *ΔA* gives information on the evolution of the carrier concentration due to mono-, bi- and trimolecular recombinations, depending on the processes taking place [34,35]. Neither charge carrier trapping [36] nor geminate (or non-geminate) electron-hole recombination should occur on this timescale considering the moderate excitation fluence. From the initial number of electron-hole pairs created per NC volume (i.e., electron-hole pair density), 5.6 × 10^17^ and 5.6 × 10^18^ cm^−3^ at excitation fluences of 6 and 60 µJ/cm^2^ respectively, we can calculate the average distance between these pairs to be about 8.5 and 4 nm, respectively. These values are consistent with a fast, three-body Auger recombination [34]. Thus, we attribute the further diminution of the high energy tail taking place on the time scale of tens of ps to the effect of the non-radiative Auger recombination (i.e., Auger re-heating), as previously discussed for hybrid perovskite nanocrystals [10,12,13].

In agreement with this interpretation, the evaluated Auger recombination time constants *τ*_2,m_ or *τ*_1,l_ decreased with the excitation fluence (see Table 3). We note that the Auger time constants *τ*_1,l_ extracted from the analysis of the long time range are about twice as large as the value *τ*_2,m_ extracted from the middle time range. This shows that the non-radiative Auger recombination rate is time-dependent and cannot be assigned a well-defined time constant. Indeed, rather than mono-molecular dynamics leading to single exponential decay, Auger recombination in weakly confined NCs is a tri-molecular process which effectively leads to highly time-dependent exponential decays. That is why the time constants in the tens of picosecond time range are correlated with the ones in the hundred of picosecond time range where the high-energy tail is still evolving but in a much reduced manner. This also leads us to assign the longer time constants *τ*_3,m_ and possibly *τ*_2,l_ to the end of the Auger recombination process. Finally, at the longest measured times, the non-geminate electron-hole recombination leads to the disappearance of the bandgap bleach following bimolecular kinetics not related to cooling dynamics. This thus might partially affect *τ*_3,m_ and *τ*_2,l_, in addition to the *τ*_3,l_ time component in several nanoseconds assigned to this recombination.

## 5. Conclusions

In conclusion, we have investigated the hot charge carrier cooling dynamics of colloidal FAPbI_3_ nanostructures in the weak confinement regime. Overall, these results show that the evolution of the lineshape and signal amplitude in time as obtained from global TA data analysis allows us to disentangle the different processes behind the charge carrier relaxation and recombination in these samples. The extracted kinetic parameters over short times (the first few picoseconds) give a good description of the main cooling dynamics through LO-phonon emission, with an important hot phonon bottleneck effect slowing down the relaxation to almost 1 ps for a charge carrier density of about 6 × 10^18^ cm^−3^. While more sophisticated analysis going beyond a simple exponential behavior could be applied over longer times, the sequential kinetic model used successfully shows the involvement of Auger re-heating in both cooling and recombination mechanisms from a few tens to hundreds of picoseconds, for a charge carrier density of 10^18^–10^19^ cm^−3^. Finally, we emphasize the importance of employing the global analysis method, without which independent observations of the time-evolution of the line-shape and the amplitude of the very complex TA spectra would have been impossible.

## Figures and Tables

**Figure 1 nanomaterials-10-01897-f001:**
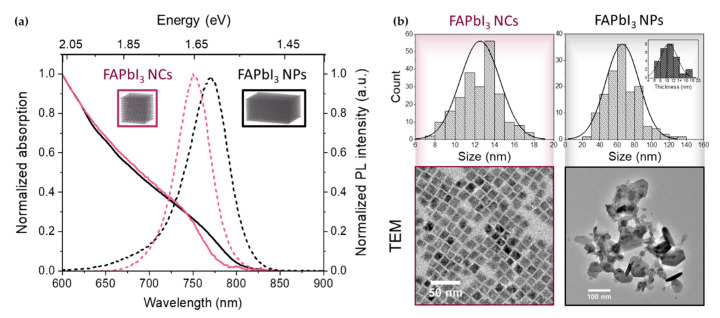
(**a**) Absorption (solid line) and photoluminescence (dashed line) spectra of the corresponding colloidal FAPbI_3_ samples (nanocrystals (NCs) in purple, thick nanoplates (NPs) in black). (**b**) TEM characterization of the FAPbI_3_ NCs and thick NPs. Inset: histogram corresponding to the NP thickness.

**Figure 2 nanomaterials-10-01897-f002:**
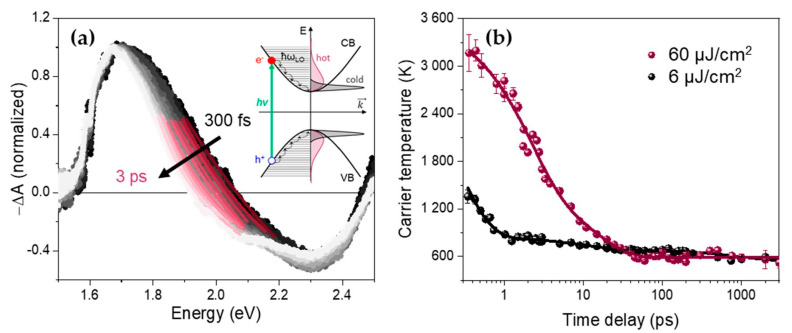
(**a**) Normalized transient absorption (TA) spectra of thick FAPbI_3_ NPs during the first 3 ps (dots, excitation 630 nm, 160 μJ/cm^2^, chirp-corrected, see Appendix A). The high-energy tail fitting is shown in wine-colored solid lines. Inset: schematics of the relaxation from a hot to cold charge carrier distribution in the case of a continuum of energy levels above the bandgap. (**b**) Time-dependent carrier temperatures *T_c_* of FAPbI_3_ NPs extracted over five decades of time from the TA spectra for a 630 nm pump excitation at 6 μJ/cm^2^ (black dots) and 60 μJ/cm^2^ (wine-colored dots) using the conventional tail-fitting method. The multiexponential fits of these decays are displayed with solid lines (parameters in Table 1).

**Figure 3 nanomaterials-10-01897-f003:**
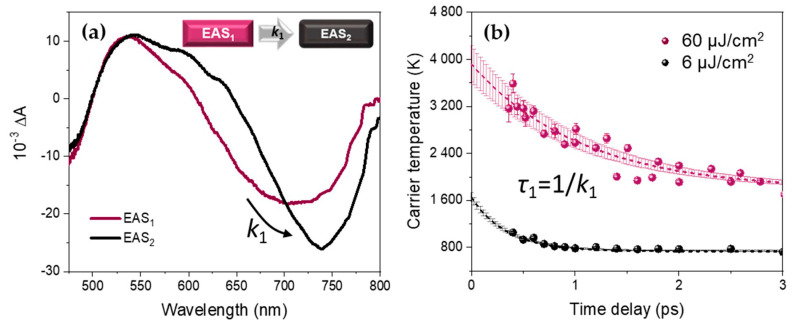
(**a**) Evolution Associated Spectra EAS_1_ and EAS_2_ related by the relaxation constant *k*_1_ extracted by the global analysis from the TA data of thick FAPbI_3_ NPs (here for an excitation fluence of 60 µJ/cm^2^). (**b**) Comparison between *T_c_* extracted from the conventional analysis (dots) and the exponential simulation using global fit parameters (dashed lines with the associated incertitude in the shaded area) at 6 µJ/cm^2^ (black) and 60 µJ/cm^2^ (purple) excitation fluence, using Equation (4).

**Figure 4 nanomaterials-10-01897-f004:**
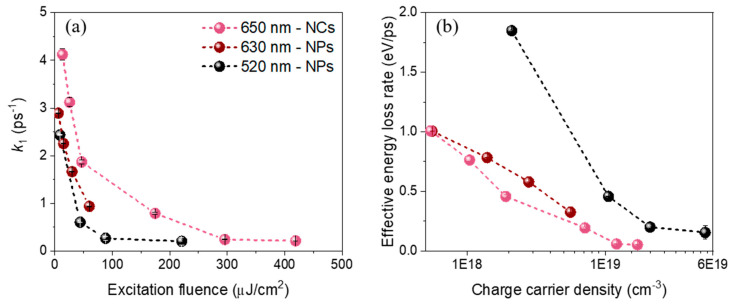
(**a**) Extracted cooling rates *k*_1_ of FAPbI_3_ NCs (pink dots, excitation 650 nm) and thick NPs (excitation 630 nm: wine-colored dots, excitation 520 nm: black dots) plotted versus the excitation fluence. (**b**) Plot of the corresponding energy loss rate versus the charge carrier density (number of initial electron-hole pairs per nanocrystal volume).

**Figure 5 nanomaterials-10-01897-f005:**
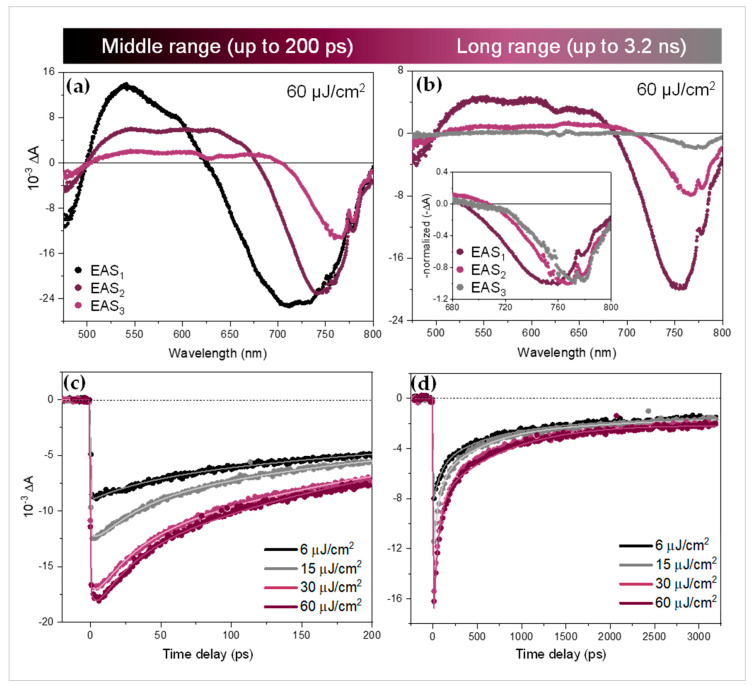
EAS of thick FAPbI_3_ NPs excited at 630 nm for the (**a**) middle and (**b**) long time ranges at 60 µJ/cm^2^. Inset: normalized EAS. (**c**) Experimental TA decay traces (dots) and fitted traces (lines, from global analysis) of thick FAPbI_3_ nanoplates detected at the band-edge (765 nm) for the middle and (**d**) long time ranges at several excitation fluences. The fit decays can be simulated based on Equations (2) and (3), using the corresponding EAS (for example in Figure 5a,b or Figure A2 in Appendix B) and linked kinetic constants displayed in Table 3).

**Table 1 nanomaterials-10-01897-t001:** Fit parameters for a tri-exponential decay of the charge carrier temperature (*T_c_*) at moderate and high excitation fluence. We use the following equation: Tc=T0+∑i=1nai exp(−tτi) and $A_{i}=\frac{100\;a_{i}}{\sum_{j}a_{j}}$.

Excitation Fluence (µJ/cm^2^)	*A*_1_ (%)	*τ*_1_ (ps)	*A*_2_ (%)	*τ*_2_ (ps)	*A*_3_ (%)	*τ*_3_ (ps)	*T*_0_ (K)
6	90 ± 30	0.25 ± 0.05	5.7 ± 0.5	11 ± 2	5 ± 2	1030 ± 860	540 ± 60
60	70 ± 4	2.1 ± 0.6	30 ± 4	14 ± 2	-	-	588 ± 8

**Table 2 nanomaterials-10-01897-t002:** Fit parameters for the simulation of charge carrier temperature (*T_c_*) and the extracted cooling times of thick FAPbI_3_ NPs obtained from global analysis at different excitation fluences.

Excitation Fluence (µJ/cm^2^)	*T*_i_ (K)	*T*_f_ (K)	*τ*_1_ (ps)
6	1648 ± 71	731 ± 6	0.36 ± 0.02
15	1919 ± 82	937 ± 8	0.44 ± 0.02
30	2931 ± 25	1399 ± 15	0.62 ± 0.02
60	3920 ± 30	1770 ± 22	0.97 ± 0.08

**Table 3 nanomaterials-10-01897-t003:** Fit parameters for a tri-exponential decay extracted from global analysis at different excitation fluences.

Excitation Fluence (µJ/cm^2^)	*τ*_1,m_ = 1/*k*_1,m_ (ps)	*τ*_2,m_ = 1/*k*_2,m_ (ps)	*τ*_3,m_ = 1/*k*_3,m_ (ps)	*τ*_1,l_ = 1/*k*_1,l_ (ps)	*τ*_2,l_ = 1/*k*_2,l_ (ps)	*τ*_3,l_ = 1/*k*_3,l_ (ns)
6	0.3 ± 1	42.9 ± 0.3	510 ± 10	82 ± 1	600 ± 10	[>3 ns]
15	5 ± 1	34.5 ± 0.2	400 ± 10	63.0 ± 0.6	520 ± 10	[>3 ns]
30	4 ± 1	29.4 ± 0.1	320 ± 10	58.8 ± 0.3	640 ± 10	[>3 ns]
60	4 ± 1	25.1 ± 0.1	280 ± 10	58.9 ± 0.4	810 ± 10	[>3 ns]

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
