# Peer review of "Charge Carrier Relaxation in Colloidal FAPbI3 Nanostructures Using Global Analysis"

_nanomaterials, 2020, doi:10.3390/nano10101897_

Round 1

Reviewer 1 Report

I read the manuscript "Charge carrier relaxation in colloidal FAPbI3 nanostructures using global analysis". This contains solid experimental results, but does not seem accessible to general readers. I would like to point out the followings before recommending publication.

  1. The authors should show a figure to explain pump-probe setups (delay stage, sample, detector...). It is now hard to see whether the authors measure transmission or reflection.
  2. What is the value of the band gap? This should be discussed with the pump wavelength. Why is it 630 nm and 520 nm for NPs and 650 nm for Ncs? 
  3. In Fig. 2 (b), a carrier temperature of 3000 K was obtained. Is it also consistent with the rough estimation from pump fluence and specific heat?
  4. How are the delay scans in Fig. 5 (c,d) fitted? The authors should show the fitting equation, including how to obtain the values in Table 3?
  5. Now Ref. 7 = Ref. 12, and Ref. 8 = Ref. 13. Refs. 12-13 should be removed. 

Author Response

We thank the Reviewer for the suggestions.

1. We added some technical details about the specificities of our TA setup in the Supplementary Materials. However, the figure of the full setup was already submited to another journal and thus is not provided here. We specified that the measurements were made in transmission in both Supplementary Materials and in the main text.

2. We added the value of the bandgap energy in the description of the materials. We choose a pump at 520 and 630 nm to excite the thick NP sample in order to analyze the effect of the excess energy on the cooling process. These two values correspond to the minimum and maximum wavelength of our NOPA source where a large range of excitation fluences can be obtained (full range of the TA setup between 500 nm and over 650 nm). The experiments on the cubic-shaped NCs were performed sometimes prior those conducted in the NPs and we added for a rough comparison to the 630 nm excitation of the NPs. We agree that it does not match exactly in terms of excess energy relative to the bandgap.

3. The maximum carrier temperature is given by the excess excitation energy relative to the bandgap ΔE. For the thick NP sample excited at 630 nm, we obtained Tc,max = ΔE/kB = 4038 K. At the larger excitation fluence, we found an initial carrier temperature (extrapolation of the exponential fit at time zero) of 3537 K, which is consistent with the maximum calculated value of Tc.

4. The fits shown in Figure 5 (c,d) are obtained from the global analysis using Glotaran. These fits can be recover using Equations 2 and 3 as described in the Materials and Methods section, using the corresponding EAS spectra (for example Figure 5a,b and Figure A2 in Appendix B) and parameters given in Table 3. More details about global analysis using Glotaran software can be founded in the references 25-26.

5. The repeated references were removed.

Reviewer 2 Report

In the manuscript (nanomaterials-908626) titled “Charge carrier relaxation in colloidal FAPbI3  nanostructures using global analysis”, the authors propose to employ a global analysis based on the sequential kinetic method to model the transient absorption of FAPbI3 nanostructures and compare it with the conventional method based on the Maxwell-Boltzmann function.

The manuscript is well structured and well written. The final discussion is interesting and well argued. Therefore, in my opinion, the manuscript is worth publishing.

However, some revisions are required:

  • Fig1a, Please, insert on the top axis the energy values (eV)
  • Fig3b, It seems that the Tc data extracted by using the conventional analysis are the same of those in Fig.2b. The first point in Fig.2b is at 300fs, while in Fig. 3b there are dots at lower delay times. Please, clarify if the data (dots) are the same and consequently why these dots are missing in Fig.2b
  • The authors reproduce the middle and the long range transient absorption data by using three kinetic components for each range. The second time constant of the middle range is few tens of ps (25-40 ps) and corresponds to EAS2 to EAS3 decay. Is this comparable with the relaxation from EAS1 to EAS2 (time constant about 60-80 ps) of the long range? And the decay of EAS3 in the middle range is comparable with EAS2 to EAS3 decay of the long one? To avoid confusion, I suggest to number the EAS for both ranges in a sequential mode and to attribute the same number to the comparable EAS. I suggest also to pay attention to the error bars in order that the EAS are really comparable.

After these revisions, I can recommend the publication in Nanomaterials.

Author Response

We thank the Reviewer for his suggestions.

- Fig1a. We added the scale in eV.

- Fig3b. There are the same data points. Carrier temperatures should normally not be extracted at early time (below 300 fs), during the thermalization stage. We thus excluded also the first one or two points in Figure 3b as well.

- In the paper, we assigned the second step of delayed relaxation to Auger reheating effects. This second step is both between EAS2 and EAS3 in the middle range experiment and EAS1 to EAS2 in the long range as written in the manuscript: "This can be observed by comparing EAS2 to EAS3 in the middle range experiments associated with time constant τ2,m (Figure 5a) or EAS1 to EAS2 in the long time range data associated with the τ1,l (Figure 5b)." We then in the discussion explain why we found a factor 2 between the two characteristic times: "In agreement with this interpretation, the evaluated Auger recombination time constants τ2,m or τ1,l decrease with the excitation fluence (see Table 3). We note that the Auger time constant τ1,l extracted from the analysis at long time range are about twice as large as the value τ2,m extracted at middle time range. This shows that the non-radiative Auger recombination rate is time-dependent and cannot be assigned a well-defined time constant. Indeed, rather than a mono-molecular dynamics leading to single exponential decay, Auger recombination in weakly confined NCs is a tri-molecular process which effectively leads to highly time-dependent exponential decays. That is why the time constants in the tens of picosecond time range are correlated with the ones in the hundred picosecond time range where the high-energy tail is still evolving but in a much reduced manner. This also leads us to assign the longer time constants τ3,m and possible τ2,l to the end of the Auger recombination process." The difference between the values are much larger than the associated incertitudes.

Reviewer 3 Report

Villamil Franco and colleagues empoy transient absorption to study relaxation processes in perovskites. They find out that, at high fluency, the relaxation dynamics is slower and attribute this to a "hot photon" bottleneck.
The paper is in general well written, the experimental design is appropriate and the conclusions are supported by the results. For this reasons, I support it for publication in "Nanomaterials" provided that the minor points listed here below are addressed.

Line 159: there are two corrections left.
Line 187: <100 > --> <100>
Line 222: "These curves can be fitted satisfactory" It'd be good to quantify what "satisfactory" means by writing a chi squared, and R-factor or another goodness-of-fit index. This is important also to support the statement that "A larger number of spectral/kinetic components did not result in a significant improvement of the fit."

Author Response

We thank the Reviewer for his comments.

L159. The corrections were removed.

L187. The space was removed.

L222. The fit quality is qualitatively observed in the Figure 2b.  For a quantitative analysis, the coefficient of determination R2 for the 60 µJ/cm2 temperature decay curve fitted by a bi-exponential function (fit from Origin program) is R2=0.985. For the 6 µJ/cm2 temperature decay curve, R2=0.933, using 3 exponentials. The values are added to the main article.

The fit of the TA data in time and wavelength using global analysis from Glotaran program were obtained using two components (2k) at short time range and three components (3k) at medium and long time range. The quality of the fit can be observed in the selected decay traces of Figure A1c,d in Appendix A and Figure 5c,d. For the quantitative quality of the fit, we used the root mean square deviation given by Glotaran and found no improvement between 2 and 3k in the fit of the TA data at short time range, while a non-negligeable improvement was found for the fits at medium and long time range. The text was modified accordingly.

Round 2

Reviewer 1 Report

I read the revised manuscript "Charge carrier relaxation in colloidal FAPbI3 nanostructures using global analysis". Now I am satisfied with the revised manuscript and Supplementary Materials, and recommend publication in the present form.